# Association between Oral Cancer and Diet: An Update

**DOI:** 10.3390/nu13041299

**Published:** 2021-04-15

**Authors:** Jesús Rodríguez-Molinero, Blanca del Carmen Migueláñez-Medrán, Cristina Puente-Gutiérrez, Esther Delgado-Somolinos, Carmen Martín Carreras-Presas, Javier Fernández-Farhall, Antonio Francisco López-Sánchez

**Affiliations:** 1Department of Nursing and Stomatology, Faculty of Health Sciences, Rey Juan Carlos University, 28922 Alcorcón, Madrid, Spain; blancac.miguelanez@urjc.es (B.d.C.M.-M.); cristina.puente@clinica.urjc.es (C.P.-G.); esther.delgado@urjc.es (E.D.-S.); antonio.lopez@urjc.es (A.F.L.-S.); 2Adult’s Dentistry Department, Oral Medicine, Universidad Europea, 28670 Villaviciosa de Odón, Madrid, Spain; carmen.martin2@universidadeuropea.es; 3Faculty of Health Sciences, Rey Juan Carlos University, 28922 Alcorcón, Madrid, Spain; perioim@yahoo.es

**Keywords:** oral cancer, diet, proinflammatory diet, cancer prevention

## Abstract

Oral cancer, included within head and neck cancer, is the sixth most common malignant neoplasm in the world. The main etiological factors are tobacco and alcohol, although currently, diet is considered an important determinant for its development. Several dietary nutrients have specific mechanisms of action, contributing to both protection against cancer and increasing the risk for development, growth, and spread. Foods such as fruits, vegetables, curcumin, and green tea can reduce the risk of oral cancer, while the so-called pro-inflammatory diet, rich in red meat and fried foods, can enhance the risk of occurrence. Dietary factors with a protective effect show different mechanisms that complement and overlap with antioxidant, anti-inflammatory, anti-angiogenic, and anti-proliferative effects. The main limitation of in vivo studies is the complexity of isolating the effects related to each one of the nutrients and the relationship with other possible etiological mechanisms. On the contrary, in vitro studies allow determining the specific mechanisms of action of some of the dietary compounds. In conclusion, and despite research limitations, the beneficial effects of a diet rich in vegetables and fruits are attributed to different micronutrients that are also found in fish and animal products. These compounds show antioxidant, anti-inflammatory, anti-angiogenic, and anti-proliferative properties that have a preventive role in the development of oral and other types of cancer.

## 1. Introduction

The term “head and neck cancer” includes malignant tumors with an upper aerodigestive tract location: oro-pharynx, larynx, and hypopharynx [1]. Globally, it is the sixth most common malignant neoplasm, with an especially high prevalence in Southeast Asia [2]. Oral cancer specifically includes a subgroup of neoplasms arising in lips, the anterior two-thirds of the tongue, gingivae, hard and soft palate, oral mucosal surfaces, and floor of the mouth [3]. Of these oral cancers, more than 90% are oral squamous cell carcinomas (OSCC) [4,5,6].

Annually, approximately 355,000 oral cancer cases and 93,000 oropharyngeal cancer cases are diagnosed, representing 2% and 0.5% respectively of the malignant neoplasms detected worldwide [7]. The incidence is higher in men than in women (5.8 and 2.3 per 100,000 individuals respectively); however, there seems to be an increasing trend in women mainly due to their relatively recent exposure to risk factors such as tobacco or alcohol [3]. The mean age at presentation is around the sixth decade of life, although an increase in onset has been observed in young adults (less than 45 years), mainly related to genetic predisposition and human papillomavirus (HPV) [8]. The prevalence is particularly high in Asia (64% of the total diagnosed cases), followed by Europe (17.4%) and North America (7.6%) [7]. This increased prevalence in Asian countries may be related to risk factors exposure such as smokeless tobacco and the consumption of alcohol and betel quid chewing [9]. In Europe, since the 1990s, an increase has been reported mainly due to a greater exposure to known risk factors such as alcohol and tobacco [10].

The 5-year survival rate for oral cancer is still low, despite both diagnostic and therapeutic progress [11].

The survival rate varies depending on factors such as anatomical location and tumor stage at the time of diagnosis. If the tumor is diagnosed in the early stages, the survival rate is around 80–90%, thus limiting both the extent of surgical treatments and the administration of adjuvant therapies [12]. 

Oral cancer risk factors are widely known and include tobacco (chewing/smoking), alcohol, betel quid, genetic factors, socioeconomic status, poor oral hygiene, human papillomavirus (HPV), and diet [13].

The association between dietary products and the risk to develop cancer has been recently evaluated. If studies reporting on alcohol and tobacco are not included, there seems to be an increased risk in relation to certain foods or families including those rich in pro-inflammatory factors. On the other hand, certain nutrients, micronutrients, and food components can act as protective elements. This protective effect can be obtained from fruits, vegetables, and certain vitamins as well as in other foods and products common to our diet. Low consumption of fruit and vegetables has been associated with an increased risk of oral cancer. Therefore, geographic areas with poor access to these foods have higher prevalence than regions where they are frequently consumed [14].

Numerous studies seem to indicate that different food compounds could alter or modify cancer cells. Thus, different dietary products would be effective in the prevention and treatment of certain types of cancer. Specific foods would act by producing epigenetic cellular changes and modifying their genetic material. The term “epigenetics” refers to changes in gene expression and chromatic remodeling independent of the DNA sequence itself [15]. These changes typically occur by DNA methylation, histone modifications, or genetic expressions by nc-RNA [16]. Epigenetic modifications can play an important role in the development of cardiovascular diseases, autism, obesity, type-2 diabetes, and cancer [17,18,19]. DNA methylation occurs when a methyl group is added at the 5 (C5) carbon position of Cytosine Guanine (CG) dinucleotides, called CpG dinucleotides, of a cytosine. Cytosine methylation involves the transfer of a methyl group from S-adenosyl-L-methionine (SAM), which is a methyl precursor, to cytosines on CpG dinucleotides [19,20]. DNA methylation is catalyzed by enzymes known as DNA methyltransferases (DNMTs) and by the hypermethylation of CpG dinucleotides or CpG islands by DNMTs in the gene knockout transcriptional gene [16]. The human genome contains four DNMT genes, DNMT1, DNMT2, DNMT3A, and DNMT3B [21]. Alterations in the expression and activity of DNMT have been described in some of the diseases previously listed [22].

Conversely, hypomethylation is also found in virtually all human cancers [23]. Epigenetic changes create dysregulations in the transcription of genetic material, which can produce alterations in the expression or activation of factors related to oncogenes or tumor-suppressor genes [24]. Similarly, certain fruit and vegetable-rich diets, as well as rich in vitamins and minerals, could intervene in the initiation of apoptosis and the suppression of tumor genes [25,26,27,28]. Table 1.

Certain bioactive components of the diet appear to have the potential to prevent different types of cancer, including oral cancer [29,30]. These micronutrient bioactive compounds are present in dietary elements such as vegetables, fruits, teas, garlic, and cereals, among others. Different cellular DNA regulation mechanisms have been suggested [16].

**Table 1 nutrients-13-01299-t001:** Bioactive epigenetic diet compounds, food sources, epigenetic functions, anti-inflammatory effects and cancer types related. Modified from: Hardy, T.M.; Tollefsbol, T.O. Epigenetic Diet: Impact on the Epigenome and Cancer. Epigenomics 2011, 3, 503–518. DNMT: DNA methyltransferase; EC: Epicatechin; ECG: Epicatechin-3-gallate; EGC: Epigallocatechin; EGCG: Epigallocatechin-3-gallate; HAT: Histone acetyltransferase; HDAC: Histone deacetylase.

Epigenetic Diet Compounds		Food Sources	Epigenetic Functions	Anti-Inflammatory Effects	Other Effects	Related Cancers
	Epigallocatechin gallate (EGCG) [28,31]	Green tea	DNMT and HAT inhibitor, modulates miRNA			Oral, breast, prostate, gastric, ovarian, esophageal, skin, colorectal, pancreatic, and head and neck
	Resveratrol [28,32,33]	Grapes, peanuts, mulberries, cranberries, Blueberries	DNMT and HDAC inhibitor		Antioxidant activity	Liver, skin, breast, prostate, lung and colon
	Curcumin [28,34]	Turmeric, curry	DNMT inhibitor and miRNA modulator	Halogenated cytosine products (mimic 5-methylcytosine in DNA methylation)		Leukemia, hepatoma, cervical and pancreatic
Flavonoids	Genistein [28,35,36,37]	Soybeans, fava beans	DNMT and HDAC inhibitor, enhances HATs, modulates miRNA			Cervical, prostate, colon and esophageal
Isothiocyanates [28]		Broccoli, cabbage, kale, watercress	DNMT and HDAC inhibitor			Myeloma, leukemia, colorectal and prostate
Omega 3 acids [38]		Flaxseed and soybean oil, tuna, salmon, mackerel and other seafood		Inhibitory effect on arachidonic acid synthesized eicosanoids		Prostate and breast
Minerals	Selenium [28,39,40]	Brazilian nuts, chicken, game meat, beef	DNMT and HDAC inhibitor	Inhibits prostaglandins	Antioxidant activity	Skin
	Zinc [28,41]	Animal proteins: beef, pork and sheep	DNA repair		Antioxidant activity	Oral
	Magnesium [28,41]	Dark green leafy vegetables. Nuts, cereals, legumes and yeast	DNA repair			
Organosulfur Compounds	Allyl mercaptan [28,35,42]	Garlic	HDAC inhibitor Histone acetylation	Inhibition of lipoxygenase and cyclooxygenase	Antioxidant activity	
Vitamins	Folate [28,43,44,45,46]	Beans, grains, fortified breakfast cereals, pastas, green vegetables	Deficiencies alter DNA methylation patterns			Breast, cervix, ovary, brain, lung and colorectal
Carotenoid pigments	Lycopene [47,48,49]	Tomatoes, papayas, and watermelons			Antioxidant activity	Prostate and oral
Pro-inflammatory diet [38,50,51,52,53]		Fried meats and omega 6 acids	Nitrogenated compounds and polycyclic hydrocarbons		Arachidonic acid activity	Oral, breast, prostate, lung, gastric and colorectal
Alcohol [28,43,54]		Alcoholic beverages	High consumption increases promoter hypermethylation		Low folate absorption	

This study aims at investigating the relationship between oral cancer and diet, as well as to highlight which factors may be detrimental and which may be considered protective due to their properties. In addition, these possible relationships are important not only for the scientific community but also for the general population.

## 2. Pro-Inflammatory Diet

Dietary habits have been consistently linked to the development of several types of cancer. The presence of inflammatory serum factors such as C-reactive protein (CRP), interleukin (IL)IL-1β, IL-4, IL-6, IL-10, and tumoral necrosis factor (TNF) TNF-α have been studied in certain types of diet, establishing food that promotes inflammation and food that reduces this inflammatory condition [50]. The balance between them will determine the inflammatory potential of the diet.

A pro-inflammatory diet induces persistent inflammation, promoting cancer development in some parts of the body, including oral cancer [50]. The dietary inflammatory index (DII) has been developed to establish parameters that help analyze the dietary patterns and the risk of different types of diet. The current research indicates that the DII relates the diet and the C-reactive protein in serum to the degree of inflammation [55]. High DII values indicate pro-inflammatory diets, while low DII levels would indicate anti-inflammatory diets and a consequent lower risk to develop oral cancer [50].

There are several ways in which a pro-inflammatory diet can increase the risk for oral cancer: first of all, through the production of biomarkers such as CRP, IL-6, and homocysteine. Thus, the inflammatory process is responsible for providing bioactive molecules to the tumor environment. In addition, the inflammatory transcription factors can also be activated by cytokines and other inflammatory biomarkers, intervening in the initiation and promotion of cancer. Modification of the dysbiotic oral microbiota has been proposed as another possible route, establishing a possible association with head and neck cancer [56].

Among the pro-inflammatory factors, a high intake of iron has been associated with oral squamous cell carcinoma (OSCC), and it is similarly shown in other tumors such as lung, prostate, and breast cancer. This could be explained because iron participates in fundamental cellular functions, such as cell metabolism, growth, and proliferation, which can lead to the production of nitrogen compounds and catalyze the formation of free radicals that cause cell damage [57].

In addition to iron, natural red meat contains other components such as nitrates and nitrites that can contribute to the development of oral cancer. Furthermore, when cooked, other carcinogenic mechanism, such as heterocyclic amines and polycyclic hydrocarbons production are generated [58].

The potential carcinogenic effect of nitrates is a consequence of their conversion to methemoglobin-producing nitrite. This methemoglobin cannot bind oxygen and can cause hypoxemia. When they are kept fresh, the concentration of nitrite (e.g., in vegetables) is usually low, but depending on the storage conditions, nitrite concentrations may increase either by bacterial contamination or by endogenous nitrate reductase. If foods are stored in a refrigerator, nitrate reductase is inactivated [51,52]. 

These compounds are also found in cruciferous vegetables, in low or average concentrations; however, because they are consumed frequently and in large quantities, they can become a significant source in the daily diet; although these levels can be reduced depending on the cooking technique [53]. To reduce the amount of nitrate intake, nitrogen fertilizers should be reduced, and correct storage and processing should be ensured to reduce the risk of bacterial contamination [51,52].

Omega 6 acid has also been linked to OSCC patients. The mechanism of action is based on the metabolism of Omega 6 acid producing arachidonic acid that generates pro-inflammatory prostaglandins and lipoxins by oxidation. The balance between Omega 6 and Omega 3 acids can regulate the action of carcinogenic factors and decrease the risk of oral cancer [38].

Fried foods have been directly linked to stomach, rectal, and colon cancer. Research has been conducted to elucidate if this also exists in oral cancer, concluding that a moderately increased risk of oropharyngeal carcinoma is observed in men with a diet rich in fried foods [58].

Other authors have analyzed the association of these pro-inflammatory foods with laryngeal cancer. This relationship was found to be stronger in individuals who were overweight [59].

Other dietary products with high glycemic potential, such as soft drinks and desserts, rapidly increase blood glycemic indices, leading to a rise in insulin plasma levels —which is a hormone related to tumor proliferation, although there is no specific data that link these foods to oral cancer [59].

## 3. Protective Diet

For decades, there have been certain trends referring to a “green chemo-prevention”, suggesting a preventive aspect of a healthy lifestyle and a diet based on natural foods with a well-known protective “role”, as a preventive philosophy approach toward cancer [60].

In 1997, the World Cancer Research Fund International and the American Institute for Cancer Research published a consensus report establishing scientific evidence associating high consumption of fruits and vegetables with a lower risk of suffering oral, pharyngeal, esophageal, lung, colon, and rectal cancer [61]. This has further been confirmed in several studies where a protective relation against the most frequent head and neck cancer (OSCC) was reported. Furthermore, a dose-dependent relation was mentioned, where a higher intake of these foods was related to a lower risk of developing previously mentioned cancers [59,62]. Some mechanisms have been suggested to explain the possible protective effects. Antioxidants decrease reactive oxygen species. Some compounds found in vegetables are thought to have anti-tumoral properties, such as glycates and indol-3 carbonol (inducing phase II enzymes) responsible for eliminating reactive oxygen species and DNA repair [63].

Vitamins found in numerous dietary products have antioxidant and anti-proliferative properties, including immune system enhancement with synthesis and DNA methylation [64].

Vitamin C intake, as found in citrus fruits, reduces the risk of developing primary cancers, although no clear evidence has been reported in relation to secondary cancers. No recommended dose has been established, but in any case, the risk reduction seems clear [65]. Vitamin C has a reducing effect in the cell. There seems to be a synergistic effect of vitamin C and vitamin E (in charge of eliminating free radicals in the cell membrane). Vitamin C also protects against nitrosamine production and the union between DNA and certain carcinogens, leading to chromosomic damage, thus lowering the risk of developing cancer though different mechanisms [66,67,68].

In relation to dark yellow fruits (e.g., oranges, lemons, apricots), no association has been established between their consumption and a lower risk of oral, pharyngeal, or esophageal cancer [69]. Nevertheless, in a study performed in Brazil, the authors observed that banana consumption reduced the risk of head and neck cancer diagnosis by 77%. This fruit contains vitamins, phenolic acids, carotenoids, and biogenic amines with an antioxidant effect [70].

Red fruits such as blackberries and redberries, along with grapes, contain large quantities of polyphenols such as resveratrol [32]. This compound has anti-inflammatory, antioxidant, and anti-cancerous properties. Resveratrol controls cell growth, cell division, cell migration, cell adhesion, and cell invasion, along with apoptosis and angiogenesis [33].

Vegetables contain high levels of micronutrients (beta-carotene, alpha-carotene, lycopene, vitamins A, C, and E) with anti-cancerous properties and, in some cases, the combination of several molecules enhances their properties [69]. Beta-carotene are antioxidants, protecting from DNA damage. The transformation into retinol plays a key role as this molecule participates in cell adhesion and differentiation and membrane permeability, including a protective role against cancer.

Lycopene is a natural pigment synthesized by plants and certain micro-organisms [71]. Although it is found in fruits such as watermelons and grapefruits, ripe tomato seems to be the main source of lycopene. This compound has great antioxidant properties and has been studied for the prevention and treatment of chronic diseases such as degenerative diseases, bone disorders, and cardiovascular diseases [47]. Thus, it could be beneficial in the treatment of potentially malignant oral conditions [48,49,72,73], and as a protective factor versus oral cancer due to the regulation of lipid peroxidation and reduced glutathione (GSH) [74].

Other vegetables such as garlic, from the liliaceae family (“allium vegetables”), despite being used as a condiment, are also well known for their therapeutic properties, such as antioxidant, anti-carcinogenic, anti-inflammatory, and antimicrobial [35,42]. Garlic has a high content of organosulfur and flavonoid components that provide the flavor, but it also contains non-sulfur components that act synergistically to provide beneficial effects [75]. Studies investigating the anti-cancerous properties of garlic hypothesize that certain phytochemicals could increase the activity of enzyme systems by detoxifying carcinogens [36,76]. Some authors stress the credible evidence in relation to the association of garlic consumption and esophagus, prostate, larynx, colon, ovarian, kidney, and oral cancer [35]. This association has been studied using the available scientific evidence [36] (several methodological deficiencies are noted). Only one case-control study showed sufficient methodological quality to claim that high garlic consumption is positively associated with a reduced risk of developing these cancers [37]. Nevertheless, it is difficult to analyze the amount of garlic consumption using questionnaire data collection methods. In addition, there are variables that can condition the data such as the preparation method (raw or cooked) and the cultivation conditions. In relation to oral cancer, and with only one valid study, it can be admitted that there is very limited evidence to support the consumption of garlic and its relationship as a protective factor against cancer [36].

As previously mentioned, the combined consumption of fruits and vegetables reduces the risk of suffering from any type of cancer, stating the minimum ingested amount of 550–600 g/d, although it varies according to which foods and even between the different studies published [77]. Other authors argue that a preventive diet for cancer should include the consumption of 10 vegetables a day, in their different forms, especially raw, and also included in juices [78].

The aforementioned micronutrients are studied in depth, taking into consideration their main source and mechanism of action.

### 3.1. Folate

Folic acid or folate, also known as vitamin B_9_, can be found in vegetables, beans, cereals, and pasta. Additionally, it can be derived from both plant and animal foods (natural folate) and from supplements also known as folic acid [43]. It is an essential element in DNA methylation and has been linked to different tumors such as breast, ovary, cervix, lung, and colon cancer. Alcohol and tobacco consumption are reported to down-regulate folate levels [44,79].

Folate present in the diet has polyglutamate side chains that need to be oxidized and hydrolyzed for absorption. To enhance bio-availability, folate can be found as an oxidized pteroylglutamic acid [45].

Folate is essential for DNA synthesis, methylation, and in cell cycle repair mechanisms, modulating the risk of developing oral cancer, since the epithelium is in continuous proliferation and regeneration [46].

This participation in DNA repair suggests that folate deficiency can contribute to the development of certain types of cancer, although this association is not entirely clear. Studies have reported an inversely proportional relationship between high levels of folate and the occurrence of oropharyngeal cancer (regardless of the form of intake/presentation), although this inverse association seems to be stronger for oral cancer [80,81]. This higher risk is more prevalent in cases of heavy drinkers with low folate levels [43].

Similarly, alcohol interferes with the correct transport and metabolism of folate, increasing the risk of cancer, since it alters the synthesis, repair, and methylation of the DNA of the oral epithelial squamous cells. Furthermore, acetaldehyde production from ethanol occurs in the oral cavity, possibly impeding the potential beneficial effects of folate [54].

### 3.2. Selenium

Selenium is a well-known mineral found in walnuts, chicken, beef, and game (bush meat). Selenium has an antioxidant and DNA repair effect together with pro-apoptotic properties, acting on the methylation of DNMT and histone deacetylase (HDAC) [82]. Although some studies suggested an inverse association between selenium intake and the risk of developing cancer, recent randomized clinical studies have reported that supplementation of this mineral can increase the risk [39]. In relation to oral cancer, high levels of serum selenium act as a protective factor when combined with a high intake of fruits and fish along with a reduction in tobacco and alcohol consumption [40].

### 3.3. Zinc and Copper

Zinc is mainly found in proteins of animal origin (beef, pork, and sheep), along with dietary products such as nuts, cereals, legumes, and yeast. Zinc in combination with copper interacts in numerous biological processes such as the elimination of free radicals through the enzymatic system. Furthermore, zinc is essential in the immune response, in DNA synthesis, and in the regulation of gene transcription; therefore, zinc alterations can affect health [41,83]. High or low serum levels of both copper and zinc may have an association with the risk of developing oral cancer; thus, these parameters should be controlled in the diet [41].

### 3.4. Starch

Starch is one of the main sources of energy in the diet as it is present in multiple foods [84,85]. Different types of starch have been classified as rapidly digesting starches (RDS), slowly digesting starches (SDS), and resistant starch [86]. RDS are processed starches, and their consumption has been associated with a risk of developing oral cancer [87]. SDS travel slowly in the small intestine, and they are present in legumes and whole grains. Finally, resistant starches are non-digestible ungelatinized starches.

Depending on the type of starch, different physiological properties affecting health and specifically oral health can be observed. However, previous reviews of starch intake on oral health presented inconclusive evidence [85,88].

The consumption of certain starchy foods, especially with the refined cereals included in the RDS, has been associated with increased risk of oral cancer [87]. There is some indication that consuming dietary fiber, whole grain, and cereals may reduce the risk of head and neck cancer [89].

Recently, a systematic review analyzing the effects of total starch intake and the replacement of RDS by SDS on oral health was published. The authors confirmed the data indicating the protective effect of SDS against RDS, but they stressed the insufficient evidence due to the limitations of the included studies. In addition to the scarce amount of studies, there are biases related to the study population and the amounts of starch consumption. This leads to believe that data regarding the association of starchy foods linked with oral cancer are of low quality and that no consistent conclusions can be drawn [85].

### 3.5. Turmeric

Curcumin (*Curcuma longa* L.) is a biologic source in the Zingiberaceae family originating in Southeast Asia. Turmeric resin is used as an orange flavoring and food-coloring agent. Curcumin, a phenolic compound, is used to flavor and color several foods [90]. It has been widely used in traditional medicine (Chinese, Hindu and Ayurvedic) to alleviate digestive problems. For centuries, it has been known to exhibit anti-inflammatory, antioxidant, antiangiogenic, and anticancer activities, along with great healing potential [91,92,93].

Turmeric contains three curcuminoids: cucurmin (2.86% of curcuminoids), demethoxycurcumin (1.47%), and bis-demethoxycurcumin (1.36%) with antioxidant properties.

Curcumin has been reported to modulate the enzymatic activity in target organs, decreasing metalloproteinases (MMP-2 and MMP-9) and thus inhibiting cancer invasiveness. It also modulates the expression of EMT markers (the transition between epithelium and mesenchyme) and induces the expression of p53. Different publications reported that the administration of curcumin for 3 months (100 mg/kg) reduced the presence of different carcinogenesis substances induced by 4-nitroquinoline 1-oxide or 4NQO (used in cancer research to produce tumors in laboratory animals). It also down-regulated cellular atypia and genes expression related to EMT, along with the reduction of other processes related to the presence of HPV-16, prevention of squamous cell carcinoma, and dysplastic lesions [34].

### 3.6. Green Tea

Tea is one of the most widely consumed beverages in the world. It originates from the *Camellia sinensis* plant, with black and green being the most popular varieties.

One hour after ingesting green tea, high concentrations of catechins and flavins can be detected in saliva, enhancing a slow release of these compounds in the oral cavity. As a result, it can be effective in preventing tooth decay and periodontal disease [31,94].

The consumption of tea and especially green tea has been reported to have an inverse association with oral cancer. Several mechanisms have been proposed to explain this relationship: green tea induces apoptosis of tumor cells in oral carcinoma, and epigallocatechin-3-gallate (EGCG) inhibits the growth and invasion of tumor cells [31].

EGCG, an epigallocatechin and gallic acid esterification, has been reported to have antioxidant, anti-inflammatory, antiangiogenic, antiproliferative, pro-apoptotic, and anti-metastatic activities. These properties may explain the effect on the development, promotion, and progression of various types of tumors [94].

In vitro studies reported green tea as a natural DNMT inhibitor, thus blocking DNA hypermethylation. Epigallocatechin-3-gallate reverses the hypermethylation of RECK (reversion-inducing cysteine-rich protein with Kazal motifs), which is a known tumor suppressor gene, and it down-regulates MMP-2 and MMP-9 [94,95].

These in vitro results were not reported in other in vivo studies suggesting individual variability. Certain individuals seem to develop a protective role from green tea against oral cancer. Certain studies have investigated the role of individual oral microbiota as a possible differentiating factor. Catechins influence bacteria leading to changes in the oral microflora (*Clostridium* down-regulates and *Bifidobacterium* and *Lactobacillus* up-regulate). Depending on the person, oral microflora bacteria can metabolize the polyphenols into functional metabolites [96]. Oral microbiota has emerged as a modifying factor of the concentration of nutrients and vitamins and may contribute to the regulation of oncogenic metabolite formation [97].

### 3.7. Mediterranean Diet

The Mediterranean diet is typical in Southern European countries bordering the Mediterranean Sea: France, Portugal, Italy Spain, Greece, Malta, and Cyprus. A specific climate is also characteristic of these locations [98].

First described by Trichopoulou and Lagiou in 1997, and based on olive oil consumption in addition to frequent ingestion of fish and seafood, vegetables fruit and cereals. Other recommendations are moderate alcohol consumption and low intake of meat and dairy products [99]. The Mediterranean Diet Pyramid enhances the consumption of local seasonal products and is respectful with the environment [100,101].

People that adhere to the Mediterranean Diet (MediD) have lower cancer incidence according to the literature [102], although literature that supports this statement is scarce [99].

Filomeno et al. analyzed data from a case-control study carried out between 1997 and 2009 in Italy and Switzerland, including 768 patients with incident, histologically confirmed cancer cases and 2078 hospital controls. Three indexes were used for the evaluation: the Mediterranean Diet Score (MDS), the Mediterranean Dietary Pattern Adherence Index (MDP), and the Mediterranean Adequacy Index (MAI). They found a strong inverse association between oral cancer risk and adherence to the Mediterranean diet suggesting a strong beneficial role of the MediD on oral cancer [103].

The results of this study are in agreement with those previously published by Bosetti et al. [104] and Samoli et al. [105]. An exhaustive narrative review was carried out by Mentella et al. in 2019, where they highlighted the benefits of the Mediterranean diet in reducing cellular oxidative and inflammatory processes and avoiding DNA damage, cell proliferation, and metastasis, thus lowering the incidence of cancer [106].

## 4. Study Limitations

The vast majority of analyzed studies in this review report a series of limitations. We consider the lack of quality studies as the main problem to be able to analyze in a detailed and specific way the association between diet and oral cancer. In most of them, oropharyngeal cancer is mentioned, but no clear distinction is made in relation to oral cancer.

Epidemiological limitations also need to be mentioned. The majority of studies use questionnaires or one-time interviews, avoiding the use of prospective data collection methods, such as diet diaries, as reported by some authors. Certain bias can be detected, which is derived from inaccurate memory of food consumption in the past. Furthermore, it is very difficult to determine the amount of the different micronutrients ingested just by conducting a retrospective survey. Along with this limitation, the same foods can have several types of micronutrients, and their combined effects can be difficult to scrutinize, since multiple variables are introduced.

The amount of food, freshness, type, and cooking time, as well as shelf life, affect the number of micronutrients ingested, leading to inaccurate results [69]. Even the same micronutrients can be ingested naturally or as a supplement, complicating the quantification and bioavailability [43].

Other studies suggest limitations in the sample size and self-selection bias with participants who have a special interest for nutritional issues and are more likely to participate. In addition, patients diagnosed with head and neck carcinomas will remember their diet with greater accuracy since in some cases, and as a consequence of the medical treatment (radiotherapy, chemotherapy, or even surgical treatment), certain foods will have produced discomfort: either mucositis or even chewing limitations.

Oral microbiome and its role in the development of oral cancer has recently been studied, including certain mechanisms describing the relationship between specific microorganisms and oral cancer. Such mechanisms include enhancing chronic inflammation, attacking the epithelial barrier generating epigenetic changes, and creating oncogenic metabolites. Oral microbiota can also compromise the role of certain micronutrients in the protection or promotion of oral cancer. Several differences between in vitro and in vivo studies have been reported.

Social desirability bias has also been reported in subjects who chose their answers to please the members of the study in which anonymous or online surveys have not been conducted [107]. The methodological limitations related to ecological studies and the possibility of an ecological fallacy cannot be excluded [108]. In this type of studies, the low construct validity is another possibility that cannot be ruled out, since not all outcome-explaining variables may have been included in the methodology [109].

Research related to race, sex, or body size is scarce. We believe that these studies may have implications for public health programs in different countries [59].

Despite the progress made in the last 30 years regarding the influence of diet on the development of oral cancer, further well-designed studies are needed to reduce the methodological limitations reported in our review.

## 5. Conclusions

Numerous epidemiological studies link the diet with the prevention of different types of cancer. The influence of different types of foods on body cells is mediated through epigenetic mechanisms altering their genetic material. These same mechanisms could act on cancer cells, altering or modifying them, being effective both in the prevention of cancer and for treatment.

The beneficial effect of a diet rich in vegetables and fruits is attributed to different micronutrients, such as polyphenols, lycopene, catechins, flavins, curcuminoid, slowly digesting starches, minerals (selenium, zinc, and copper), carotenes, vitamins (A, B, C, and E), folate and omega 3 acids. Nevertheless, some are also found in fish and animal products. These compounds show different mechanisms of action, and when combined, they may have synergistic antioxidant, anti-inflammatory, anti-angiogenic, and anti-proliferative properties.

In relation to oral cancer, research is scarce. Most of the published studies refer to oropharyngeal or upper aerodigestive tract cancer without making a clear distinction with oral cancer. There are very few studies of sufficient detail and quality that identify an association between diet and oral cancer. More studies are needed to investigate a specific relationship between oral cancer and diet.

## Data Availability

Not applicable.

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
