# Peer review of "Association between Oral Cancer and Diet: An Update"

_nutrients, 2021, doi:10.3390/nu13041299_

Round 1

Reviewer 1 Report

Dear Editor
I am pleasured to review the manuscript ID: nutrients-1163440, entitled "Association between oral cancer and diet: an update". The authors reviewed the reports, most of which were published during the past one decade, and then attempted to find the relationship between oral cancer and diet. They found that different diets may contribute to either protection against cancer or increasing the risk for development, growth and spread. Foods such as fruits, vegetables, curcumin and green tea can reduce the risk of oral cancer, while the so-called pro-inflammatory diet, rich in red meat and fried foods, can enhance the risk of occurrence. They also found that food with protective effect owned various mechanisms that complement and overlap with antioxidant, anti-inflammatory, anti-angiogenic and anti-proliferative effects. Finally the authors pointed out the main limitation of this manuscript, and emphasized the advantage of in vitro studies that help search for specific mechanisms of the dietary compounds. However. some concerns makes this review need more clarification.

Introduction:

  1. Because oral cancer is the focus of this manuscript, please remove unnecessary information about other tumor sites such as pharyngeal and laryngeal areas. Even though these subsites shared certain risk factors with oral cancer, oral cancer has its own entity in cancer development, choice of treatment modality, and prognosis. Also, it is somewhat different between Asia and western countries. This should be more addressed in the Introduction since the diet habits are different.
  2. The contexts are written with jumping thinking and not organized well. This makes reader less readable. Particularly in page 2, why authors described survival and risk factors together? Risk factors is more related to cancer development and may be associated with survival. Authors should pay more attention to the difference and more address the relationship between risk factors and survival.
  3. Table 1 is not appropriately presented. It is just adapted from one published article. What is the difference between this review and the reference for readers?

Epigenetics

  1. The mechanisms are well described, but why is epigenetic mechanism more addressed here? Is it more essential to oral cancer development than the mechanisms proposed in the protective diet and micronutrients? Please clarify it.
  2. Table 2 remained out of focus. If authors attempted to deliver the information that epigenetics mediated by food and nutrients is one of the major mechanisms to oral cancer development, please delete unnecessary other cancer types in this table. Also, reference citation should be written in this table.

Pro-inflammatory diet

  1. Most of this section are clearly written, but it could be better in more description in the relation between fried food and oral cancer, not oropharyngeal cancer. Again delete unnecessary information in other sites of head and neck cancer.

Protective diet and Micronutrient

  1. Both section seems overlapped. Most of micronutrients mentioned in the context are own protective-diet characteristics. It is better reorganized.
  2. Several mechanisms are also proposed. Again delete unnecessary information in other sites of head and neck cancer.

Reviewer 2 Report

The manuscript reviews the effect of dietary nutrients on protection against cancer or  increasing the risk for development, growth and spread of oral cancer.

The review is exhaustive , even if  think it would add  value to comment on oral cancer and oral  microbiome.

Reviewer 3 Report

Rodriguez-Mlinero and colleagues prepared the manuscript on association between oral cancer and diet. In the first part of the manuscript, the authors described cancer risk factors and epigenetic mechanisms of importance in the process of carcinogenesis. Table 2 was prepared by modifying the table in the article by Hardy and Tollesfbol, 2011. The authors added several compounds to the table and columns for anti-inflammatory activity, other effects, and tumor types. In this table, I miss the bibliographical references that would allow getting acquainted with these studies.

Lines 121-127 - The authors use a mental abbreviation when writing “Omega 6 / omega 3” instead of “omega 6 / omega 3 acids”.

As an example of a pro-cancer diet, the authors only describe a pro-inflammatory diet. Only one sentence mentioned substances that promote tumorigenesis (lines 119-120). In my opinion, the authors should describe the effect of carcinogenic substances contained in food on the development of oral cancer.

The chapter on the protective diet is well written.

I assumed that this manuscript was based on the latest research results, while over 76% (75/98) of the cited articles (75/98) were published over 5 years ago.

Reviewer 4 Report

Interesting study conducted by Rodríguez-Molinero et al. The topic is quite relevant for the scientific community as well as for the population. However, some aspected need to be addressed before it can be considered for publication in Nutrients.

The main results need to be included in the abstract (quantified results) as well as the conclusions.

Lines 57-58: This statement "The association between dietary products and cancer risk has been evaluated in numerous studies." has to be well referenced and developed in the Introduction section.

The objectives of the present study have to be clearly stated at the end of the Introduction. Why is there a need in conducting the present review? What are your goals?

Lines 119-120: “Red meat may contain compounds that promote carcinogenesis such as nitrates and nitrites, as well as heterocyclic amines and polycyclic hydrocarbons in cooked meats” – Is this exclusive from red meat products? And what about the vegetables? Are they always free from health hazards? Namely nitrates, nitrites, etc ...

I recommend further literature analysis:

Keshavarz, M., Mazloomi, S. M., & Babajafari, S. (2015). The Effect of Home Cooking Method and Refrigeration Processes on the Level of Nitrate and Nitrite In Spinach. Journal of Health Sciences & Surveillance System3(3), 88-93.

Hou, J. C., gang Jiang, C., & chen Long, Z. (2013). Nitrite level of pickled vegetables in Northeast China. Food Control29(1), 7-10.

Chan, T. Y. (2011). Vegetable-borne nitrate and nitrite and the risk of methaemoglobinaemia. Toxicology letters200(1-2), 107-108.

Hsu, J., Arcot, J., & Lee, N. A. (2009). Nitrate and nitrite quantification from cured meat and vegetables and their estimated dietary intake in Australians. Food Chemistry115(1), 334-339.

LeszczyÅ„ska, T., Filipiak-Florkiewicz, A., CieÅ›lik, E., Sikora, E., & Pisulewski, P. M. (2009). Effects of some processing methods on nitrate and nitrite changes in cruciferous vegetables. Journal of Food Composition and Analysis22(4), 315-321.

What is the role of the Mediterranean Diet? Is it protective or not? Please, analyse more literature:

Serra-Majem, L., Tomaino, L., Dernini, S., Berry, E. M., Lairon, D., Ngo de la Cruz, J., ... & Trichopoulou, A. (2020). Updating the mediterranean diet pyramid towards sustainability: Focus on environmental concerns. International Journal of Environmental Research and Public Health17(23), 8758.

Fernandez, M. L., Raheem, D., Ramos, F., Carrascosa, C., Saraiva, A., & Raposo, A. (2021). Highlights of Current Dietary Guidelines in Five Continents. International Journal of Environmental Research and Public Health18(6), 2814.

Barak, Y., & Fridman, D. (2017). Impact of Mediterranean diet on cancer: Focused literature review. Cancer Genomics-Proteomics14(6), 403-408.

Mentella, M. C., Scaldaferri, F., Ricci, C., Gasbarrini, A., & Miggiano, G. A. D. (2019). Cancer and Mediterranean diet: a review. Nutrients11(9), 2059.

Filomeno, M., Bosetti, C., Garavello, W., Levi, F., Galeone, C., Negri, E., & La Vecchia, C. (2014). The role of a Mediterranean diet on the risk of oral and pharyngeal cancer. British journal of cancer111(5), 981-986.

Round 2

Reviewer 1 Report

Dear Editor
I am pleasured to review the revised manuscript ID: nutrients-1163440, entitled "Association between oral cancer and diet: an update". The authors answered the queries and revised some statements in this version. Although there is a marked improvement somewhere, most of statements and explanations in this revised manuscript are not well informed to readers.

The problems remained in this review are as follows;

  1. Oral cancer is the main focus in this review. If there is no decent amount of solid evidence related to this focus, why could authors confirmatively make a conclusion under a lot of study limitations mentioned in the text?
  2. The response to Introduction and Epigenetics remained unclear. If authors preferred inclusion, the title of this manuscript had better change to “head and neck cancer” or “cancers”, which might correspond to the main text.
  3. The authors should address the association between epigenetics and harmful habbit and expousre mentioned in the Introduction.
  4. The numerical order of tables are not correct.
  5. The “Mediterranean diet” should be separated from green tea section
  6. About red meat, do authors intend to focus on “processed / reserved red meat” or “ natural red meat”? please clarify it
  7. The conclusion of this revised manuscript may be correct and known to a variety of cancers. However, the layout and literature used are not convincing.

Reviewer 4 Report

Regarding your answer: "Point 4 and 5: We have also included data referring to carcinogenic products found in vegetables (lines 242-253). We have added a section referring to the protective role of the Mediterranean diet, as suggested. We would like to thank you very much for the provided references, which we have included in our paper. (lines 512-541)." Not all the added references are listed at the end of the manuscript. Please, pay special attention to the lines 347 - 356.
